# Mixed-Methods Approach to Land Use Renewal Strategies in and around Abandoned Airports: The Case of Beijing Nanyuan Airport

Haoxian Cai and Wei Duan *

Department of Architecture, School of Landscape Architecture, Beijing Forestry University, Beijing 100083, China; caihaoxian@bjfu.edu.cn
*   Correspondence: duanwei@bjfu.edu.cn

**Abstract:** Urban airports are typically large infrastructures with significant cultural, economic, and ecological impacts; meanwhile, abandoned airports are common worldwide. However, there is limited knowledge regarding transformation strategies for the renewal of abandoned airports and their surrounding regions in historically and culturally rich areas. We use Beijing's Nanyuan Airport as a case study, combining the historic urban landscape approach, land use and land cover change, and counterfactual simulations of land use patterns to construct a comprehensive analytical framework. Our framework was used to analyze the long-term land use patterns of the study area, determine its value, and improve perception from a macro- and multi-perspective. We discovered that the traditional knowledge and planning systems in the study area have largely disappeared, but Nanyuan Airport's impact on the surrounding land use patterns is unique and significant. By considering the characteristics and mechanisms of land use in the study area, we aimed to find a balance point between the historical context and future potential. As such, we propose optimized recommendations with the theme of connection and development engines. Our findings supplement the planning knowledge of relevant areas and provide a springboard for interdisciplinary research in landscape planning.

**Keywords:** counterfactual analysis; LUCC; CA-Markov; urban planning; historic urban landscape; China

## 1. Introduction

Large airports, as massive constructions occupying vast amounts of land, have shaped the current urban and landscape structures and will shape future developments [1]. Over time, airport hubs have evolved from mere infrastructure to multifunctional urban nodes, achieving this by regulating much larger areas beyond their immediate territories, reflecting the interaction of global flows and local dynamics [2]. However, globally, over half of all countries' airports face the risk of closure or are already abandoned. These obsolete airports often bring problems including land waste, urban pollution, and economic losses [3]. For landscape planning departments, the replanning of these sites must consider the remaining usefulness, environmental impacts, and cultural relevance after the airport's closure. Therefore, before the empirical analysis, it is necessary to outline relevant research aimed at comprehending airport-imposed urban influences, to provide a theoretical foundation for the better planning of associated regions.

In the past twenty years, academia has mainly adopted two types of models to understand the urbanization state in the vicinity of an airport: the first, designed by scholars and business strategists, comprises simplified normative "urban models", describing airport–environment relationships via "idealized spatial patterns", such as the airport city model, aerotropolis model, and airport corridor model [4]; the other relates to regional science, particularly transportation geography, logistics, territorial and spatial development, and regional economics, where scholars try to empirically assess airports' impacts on

larger regions [2,5–7]. Additionally, some scholars have conducted descriptive research on relevant urban effects based on more practical aspects, e.g., Christian Salewski et al.'s empirical descriptive model of airport-imposed urban and landscape structural influences based on five effects [1]. However, for the redevelopment of obsolete airport sites, pre- and post-closure environmental variations must be considered. Specifically, many of the airport's past effects do not completely disappear after closure but persist, shaping the regional growth differently. Thus, planning must fully acknowledge the sites' temporality. We find that the existing theoretical models and descriptive studies have limitations in their application: the theories are overly normative or abstract; case studies examine specific airports' urban effects, but, like the theories, focusing on airports themselves, they largely ignore the airports' different impacts on the surrounding area across various stages, especially pre- and post-closure. Currently, strategies to renew obsolete airports themselves and their surroundings mainly fall into five categories: Adapt, Conserve, Convert, Develop, and Regrow [3]. Which one(s) should be chosen or combined sequentially? How should their inter-relationships be considered? What should the priorities be? These limitations may lead to development plans' overall imbalance, overlooking local knowledge systems and management traditions, failing to consider the place's relevance and characteristics, struggling to guide future growth by integrating the old and new, and impeding the sites' sustainability.

Land use and land cover change (LUCC) is a good approach to understanding the above phenomena. LUCC refers to alterations in the human exploitation and utilization of land and the surficial features of natural creations and artificial constructions. It not only represents the spatiotemporally dynamic processes of the Earth's surface, but also objectively records human activities' impacts, aiding policymakers' decisions [8,9]. Some scholars have successfully assessed airports' regional influences by analyzing LUCC changes, such as Kanokporn Swangjang, who utilized ArcGIS to study land use pattern changes around Suvarnabhumi International Airport [10], and Changsheng Xiong, who adopted a land use counterfactual simulation method to study the vicinity of Hangzhou International Airport [11]. However, we find that due to satellite technology's limitations, LUCC research's time dimensions are relatively short, and most models focus more on area changes [12–15], overlooking the study regions' inherent historical cultural value and latent driving factors. The historic urban landscape (HUL) approach can resolve LUCC's defects well by combining urban planning with heritage conservation, summarizing the evolutionary patterns embedded in historical structures to provide experience and lessons for local sustainable development. There have been many successful cases [16–18], e.g., Jun Jiang et al. utilized HUL to analyze urban morphologies and landscape features across three important historical periods in Suzhou and successfully proposed optimization suggestions [19]. Likewise, we must also recognize HUL's limitations—discussions are often vague, causing disconnections between theory and practice and conservation and development [16,20–22]. Meanwhile, the application of the protection-purposed HUL alone may impede our ability to generate reasonable judgments about the complex impacts of the renewal of obsolete airports.

We find that both LUCC and HUL have individual problems but can complement each other well. Specifically, LUCC can consider obsolete airports' and their surroundings' specific land use purposes, functions, and developmental trends, providing a data foundation for their transformation. HUL can introduce the land's historical cultural components to supply principles, directions, and effective guidance. However, few studies have combined both, especially in the assessment of the renewal of obsolete airports and their vicinities. We believe that this is mainly constrained by three factors:

(1) The lack of available historical data regarding the surrounding areas of airports. The HUL method relies strongly on historical geographic data and the related literature, but many airports are located in suburban areas where the importance of the surrounding region is often overlooked, resulting in a lack of relevant records [4,23,24]. Thus, fewer studies have used the HUL method to study the airport area, and our

study fills this research gap to some extent. Meanwhile, the acquisition of historical records from satellite remote sensing images also presents difficulties as the earliest available geographic images were recorded in the 1960s, rendering them inadequate for regions with an extensive history. Consequently, researchers frequently depend on indirect and sporadic historical data, rather than direct remote sensing data, to overcome this limitation.

(2) The isolation of culture and nature. Firstly, there is a large technological gap between the two methods used, making it difficult to combine their conclusions. In addition, there is also a lack of cooperation between social scientists specializing in urban and cultural studies and natural scientists specializing in land science and technology. "Historical urban landscapes" are often used as the background for research to demonstrate spatiotemporal scales related to scientific research; however, much research mainly focuses on specific land use issues while ignoring their inherent connections with the local culture and geographical features [9].

(3) The airport as a study area with its own characteristics. Many large airports generate their own urbanization patterns due to the unique nature of their impact [1], and the related professional knowledge involved cannot be summarized by universal rules. The research threshold is relatively high. Additionally, airport sample sizes are relatively small, and not all airports require an in-depth exploration of their historical research value. Therefore, the usage scenarios are also relatively strict.

The above factors indicate that the use of a combined method to evaluate abandoned airports and their surrounding urban areas, and the proposal of corresponding planning and updating strategies, will be limited by the quantity and quality of available data. Based on this evaluation, we attempted to combine two methods to study the Nanyuan Airport in Beijing, China, and its surrounding urban area, utilizing their synergistic effect to achieve outcomes with broader significance. On one hand, historical layering was used to identify the value of historical land use patterns, related cultural values, and motivations for the redevelopment of Nanyuan Airport. On the other hand, through studying changes in land use within the research area and using CA-Markov for land simulation, we explored the impact of Nanyuan Airport on its surroundings, as well as the reasons for the changes in land use patterns within the research area. This helped to strengthen our understanding of the interrelationships between the areas inside and outside airports. Finally, we compared the results from both methods to provide more valuable information for the development of balanced and sustainable land use strategies and policies in the research area, while also providing a platform for interdisciplinary studies in landscape planning practice. It must be acknowledged that the proposed hybrid method also has certain limitations, and it cannot fully articulate the economic, cultural, and environmental impacts of airports, as they are complex systems. However, this research's goal was to provide a reference for sites' future planning by considering airports' historical spatiotemporal effects. Hence, this paper's original contribution is the introduction of a new perspective integrating quantitative analysis and qualitative judgment, emphasizing the significance of considering airports' varied impacts on their surroundings across different developmental stages, especially pre- and post-closure. This allows us to concurrently consider obsolete airports' historical heritage and future growth, compensating for the inadequacies in existing theoretical models and case studies.

In the following, Section 2 introduces the research area and data sources briefly. Section 3 elaborates on the analytical methods employed in the study, including the historical layering analysis, the land use and land cover change detection, and the use of CA-Markov for counterfactual land simulation analysis. The findings are presented in Section 4, which enables a comprehensive understanding of the historical land use patterns and interrelationships between the airport and its surroundings. Section 5 presents relevant suggestions for the updating of the land use strategies in the Nanyuan study area based on the results of both methods. Finally, Section 6 concludes the study.

## 2. Study Area and Data

### 2.1. Study Area

Nanyuan Airport in Beijing, China's first airport [25], was formally opened in 1910 and officially closed in September 2019. Figure 1 depicts the existing condition near Nanyuan Airport. Nanyuan, where it is located, was a royal hunting park throughout the Liao, Jin, Yuan, Ming, and Qing dynasties, as well as a royal garden during the Yuan, Ming, and Qing dynasties, providing a wealth of historical study value [26,27]. The Nanyuan area, as royal land, represents a fusion of the regional landscape and cultural heritage, with a relatively unified landscape type [28], formed via historical development and the long-term management of the royal land's managers and designers [29]. At the same time, because of the Chinese government's characteristics, the public nature of the airport's infrastructure development, and its potentially strong effect on urban expansion, the area's development remains overwhelmingly government-dominated [30]. Thus, the research region has been dominated by a top-down approach to land use change from ancient times to the present, and both approaches taken in our study are therefore appropriate. Furthermore, Nanyuan Airport is an essential element of Beijing's southern central axis [31], as one of the carriers of the city's historical memory, values, and spiritual culture; thus, the planning and regeneration of Nanyuan Airport and its surrounding area is of great research value.

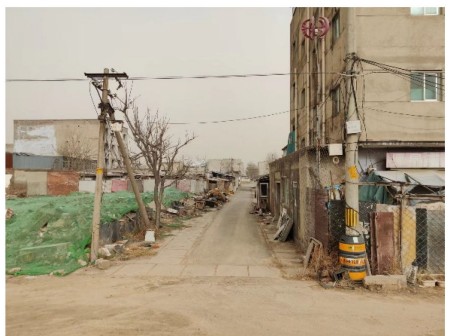 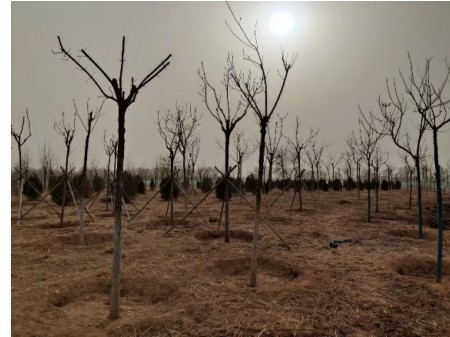

**Figure 1.** The present scenario in the vicinity of Nanyuan Airport (left photo shows the ongoing demolition and clearance work in Nanyuan, Beijing, to make way for future construction; right photo shows the protective forest area around the airport, taken by the authors in 2023).

To analyze the impact of Nanyuan Airport on the surrounding LUCC, we selected its surrounding towns as the study area, as shown in Figure 2. Due to historical reasons, the administrative divisions in the local area are relatively fragmented and scattered. We only selected certain administrative districts that were related to Nanyuan Airport. They were Nanyuan Street, Donggaodi Street, Heyi Street, and part of Nanyuan District Office, which were located in Fengtai District, as well as the Xihongmen District Office, the Old Palace District Office, the Yinghai District Office, part of the Yizhuang District Office, part of the Huangcun District Office, Guanyinsi Street, Linxiao Street, Xingfeng Street, Qingyuan Street, and the National New Media Industry Base, which were located in Daxing District.

### 2.2. Data Sources

On the one hand, using the historical urban landscape approach, we took Nanyuan as the study object, using ancient maps and regional historical accounts to determine the evolution of the historical scope of the ancient Nanyuan, the historical activities that it carried, the historical and cultural changes, the main functions that it served, and subsequently the land use patterns of different periods and the political and cultural historical connotations behind them. Specifically, we referenced the 1044 Haizi Map, the 1699 Complete Map of Nanyuan, the late Qing dynasty woodblock-brushed map of the Royal Hunting Grounds in Nanyuan, Beijing, and the 1932 Nanyuan Map to summarize changes in the historical landscape. In addition, we used official documents and data provided by the Beijing Mu-

nicipal Bureau of Natural Resources and Planning to determine the changes in the urban landscape in the Nanyuan study area from 1949 to the present. Field surveys were used to further establish the current status of the urban landscape in the Nanyuan study area.

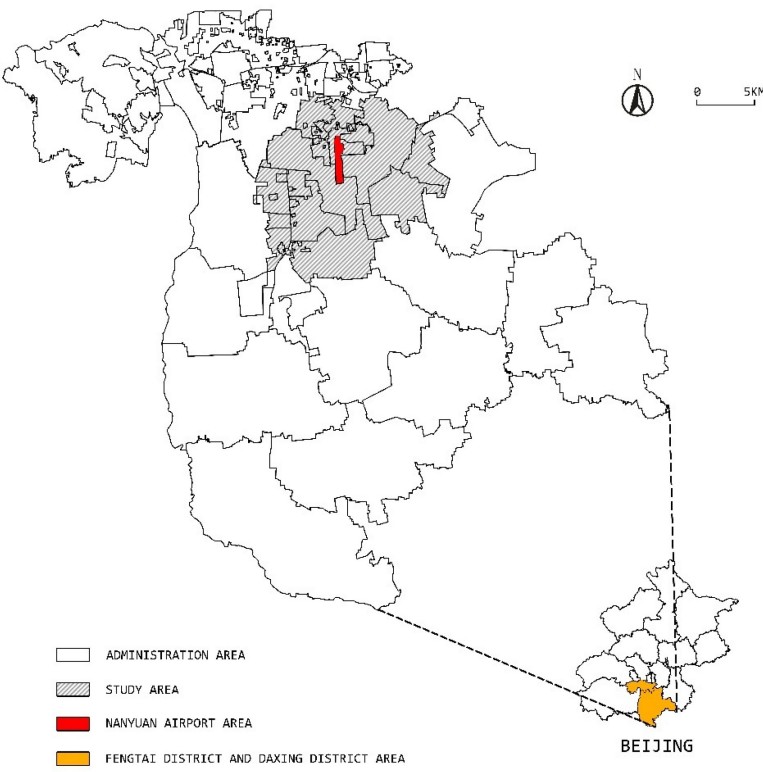

ADMINISTRATION AREA
STUDY AREA
NANYUAN AIRPORT AREA
FENGTAI DISTRICT AND DAXING DISTRICT AREA

BEIJING

**Figure 2.** Location and scope of the Nanyuan research area and Nanyuan Airport.

On the other hand, the main spatial data for the Nanyuan study area were from the USGS (https://earthexplorer.usgs.gov/ (accessed on 12 November 2022)), collecting Keyhole satellite images from 1961 and Landsat satellite data from 1984, 1993, 2001, 2011, 2019, and 2022, with cloudiness below 5% and the specific parameters shown in Table 1. The DEM data were derived from the ALOS PALSAR product data of the Alaska Satellite Facility (ASF), with accuracy of 12.5 m (https://search.asf.alaska.edu/#/ (accessed on 13 November 2022)), and from the Geographic Information Science Cloud Platform of the Computer Network Information Center, Chinese Academy of Sciences (https://www.gscloud.cn/ (accessed on 13 November 2022)), with a resolution of 30 m. The road network data were derived from OpenStreetMaps (OSM) and the LUCC data were based on Landsat satellite imagery generation. The DEM and OSM data were used to create the transition suitability image collection in CA-Markov, where the DEM data were from September 2011 and October 2019, and the OSM data were from November 2011 and September 2019.

**Table 1.** Satellite images used in the study area.

| Acquisition Date | Path | Row | Sensor Identifier | Spatial Resolution (m) | Source | Land Cloud Cover | Satellite |
|---|---|---|---|---|---|---|---|
| 30 August 1961 | / | / | / | 7.5 × 7.5 | USGS | / | KH-3 (CORONA) |
| 10 November 1984 | 123 | 032 | TM | 30 × 30 | USGS | 0.00 | Landsat 5 |
| 29 November 1993 | 123 | 032 | TM | 30 × 30 | USGS | 0.00 | Landsat 5 |
| 27 November 2001 | 123 | 032 | ETM | 30 × 30 | USGS | 0.00 | Landsat 7 |
| 23 November 2011 | 123 | 032 | ETM | 30 × 30 | USGS | 0.00 | Landsat 7 |
| 5 November 2019 | 123 | 032 | OLI_TIRS | 30 × 30 | USGS | 0.03 | Landsat 8 |
| 5 November 2022 | 123 | 032 | OLI_TIRS | 30 × 30 | USGS | 0.05 | Landsat 9 |

## 3. Methodology

### 3.1. Historical Urban Landscape Approach

The study explored the historical stratification pattern of the Nanyuan Airport study area from a HUL perspective by comparing the land use patterns and different elements of the HUL in the area from the Liao, Jin, Yuan, Ming, and Qing dynasties, as well as the Republic of China and modern China to the present day, using ancient maps and regional historical accounts. The research focused on two dimensions:

(1) Regarding the cognitive dimension, it took a macro-perspective to comprehensively examine the evolution laws of the regional land use patterns and unique historical and cultural value and to clarify the driving factors. Specifically, drawing on the World Heritage Committee's criteria for the categorization of different cultural landscapes [32,33], and combining China's cultural landscape types with the characteristics of the Nanyuan Airport area, the cultural landscape elements were divided into natural and cultural factors.

(2) Regarding the practical dimension, through the analysis of land use changes and land simulation data, it paid attention to relevant land use features and strategies and discussed methods to maintain the overall spatial pattern characteristics and highlight the multi-layered historical spatial network of the region.

### 3.2. LUCC Detection and Simulation

#### 3.2.1. Classification of Land Use Types and Testing

In order to identify alterations in land use within the study area by utilizing remote sensing imagery from multiple time periods, it was essential to maintain comparability in data acquisition and resolution. Notably, the resolution of the imagery for each year was 30 m by 30 m, with medium accuracy with sufficient bands. Therefore, to accurately classify the land use of remote sensing images, a supervised classification technique was employed [34–36]. The present study adopted the Chinese land use classification system developed by Liu Jiyuan to categorize the land use types based on the natural conditions and characteristics of the site. The system classified land use into six primary categories, namely cultivated land, forest, grassland, construction land, water, and unused land [37]. It is worth noting that there was no unused land in the study area; thus, this category was not utilized. Meanwhile, road land was added as a new land use type, and urban and rural construction land was redefined as construction land excluding roads. Additionally, we must point out the limitations in our land classification of the site. Due to data and sample constraints, reserve land shared similar features with cultivated land in the study area, making it difficult to distinguish, so we decided to merge reserve land with cultivated land. This research focused on analyzing the overall changing tendencies of regional cultivated land instead of studying specialized reserve land, so the merging was not expected to fundamentally impact the results. We will attempt to obtain higher-resolution images and reserve land samples in future studies to improve the classification accuracy. The specific classification criteria and corresponding information are presented in Table 2.

In addition, regarding the selection of training samples, we selected more than 30 evenly distributed sample points in each category, mainly by visual interpretation, with separability calculated to be greater than 1.8, allowing for classification. We then performed maximum likelihood classification using ENVI V.5.3 to classify the six types of land use. In the post-classification process, we used classification aggregation for patch aggregation. Then, we employed a mixture of visual interpretation and supervised classification results to increase the land use classification accuracy, because the human brain is adept at resolving the detection of ambiguous and nuanced spatial characteristics in visual interpretation [11]. We utilized visual interpretation in conjunction with the ENVI-classic V5.3 software to alter some of the erroneous categorization findings and modify the color scheme. Finally, we validated the classification accuracy using the confusion matrix with ground truth regions of interest (ROIs) method to compare the relationships between the ground truth points and the land cover dataset. This reflected the consistency between the current land cover

raster and ground truth [38,39]. We comprehensively evaluated the overall classification accuracy, user's accuracy, producer's accuracy, and Kappa coefficient.

**Table 2.** The land use classification and related descriptions used in this study.

| Class | Description |
|---|---|
| Cultivated land | Refers to ripe land; newly developed, reclaimed, and sorted land; recreational land (including rotational land and fallow land); land mainly planted with crops (including vegetables) or interspersed with scattered fruit trees, mulberry trees, or other trees; and reserve land. |
| Forest | Refers to land where trees, bamboo, and shrubs grow, including tree woodland, bamboo woodland, mangrove woodland, forest swamps, shrub woodland, scrub swamps, and other woodland. |
| Grassland | Refers to land where herbaceous plants grow predominantly. |
| Road | Refers to highways, urban roads, and places where social motor vehicles are allowed to pass, although they are under the jurisdiction of the unit, for the infrastructure of various trackless vehicles and pedestrians. |
| Constructed land | Refers to land where buildings and structures are built, including land for urban and rural residential and public facilities, industrial and mining land, land for transportation and water conservancy facilities, land for tourism, land for military facilities, etc. |
| Water | Refers to land that is submerged under various water bodies, such as rivers, lakes, reservoirs, canals, and wetlands, which serve as habitats for a range of aquatic organisms. |

### 3.2.2. Simulation and Counterfactual Simulation Analysis Using the CA-Markov Model

Markov chain and cellular automata modeling is one of the most common and effective methods for the modeling of the spatial and temporal changes in LUCC [40–42]. The Markov model is based on the theory of Markov random process systems formation, used for prediction and optimal control theory approaches [43,44]. The main objective of a Markov process is to determine the probability of transitioning from one state to another, wherein the future state of a random variable depends only on its current state [45]. The calculation formula for the Markov model is

$$S(t+1) = P_{ij} \times S(t) \tag{1}$$

where $S(t)$ is the system state at time $t$; $S(t+1)$ is the system state at time $t+1$; $P_{ij}$ is the transition probability matrix under the state, calculated as

$$P_{ij} = \begin{bmatrix} P_{11} & P_{12} & \cdots & P_{1n} \\ P_{21} & P_{22} & \cdots & P_{2n} \\ \cdots & \cdots & \cdots & \cdots \\ P_{n1} & P_{n2} & \cdots & P_{nn} \end{bmatrix} \tag{2}$$

$$\left( 0 \leq P_{ij} < 1 \ and \ \sum_{j=1}^{N} P_{ij} = 1, (i,j = 1,2,\ldots,n) \right)$$

where $P$ is the transition probability; $P_{ij}$ represents the probability of transitioning from current state $i$ to another state $j$ in the next time step; $P_n$ is the probability of the state at any time step.

Markov models can quantify transitions between land use types and their transfer rates, but they ignore changes in spatial patterns [46]. On the other hand, cellular automata models mainly focus on the local interactions between cells with different spatiotemporal coupling characteristics and powerful spatial computing capabilities, making them particularly suitable for the dynamic simulation and demonstration of systems with self-organizing features [44]. The formula for CA models can be expressed as

$$S(t, t+1) = f(S(t), N) \tag{3}$$

where $S$ is a finite and discrete set of cell states, $N$ is the cell field, $t$ and $t+1$ represent different times, and $f$ is the transformation rule of cell states in local space.

Cellular automata can handle the spatiotemporal dynamics of complex spatially distribution systems from the bottom up, but they have difficulties in determining changes in land use quantities accurately [47]. Considering the real-world conditions of the site and the research objectives, the CA-Markov model integrates the theories of Markov chain and cellular automata, compensating for the shortcomings of each method individually, and it was thus deemed suitable for this study. We utilized the IDRISI Selva 17.0 software to conduct CA-Markov operations, and the workflow was as shown in Figure 3. In the course of the simulations, the researchers maintained a consistent number of iterations that corresponded to the time interval of the research cycle. To establish the neighborhood definition under the cellular automata (CA), a standard $5 \times 5$ contiguity filter was employed. In order to assess the accuracy of the simulations, we employed the Kappa agreement index (KAI) approach [48], as well as the overall comparison method [49] for joint validation.

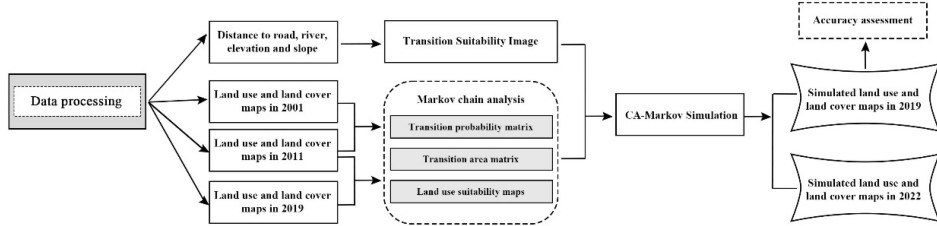

**Figure 3.** Flowchart of the cellular automata (CA)-Markov simulation workflow.

At the same time, for the purposes of this study, we adopted the idea of counterfactual simulation. A counterfactual is an 'if' statement, usually about the past [50,51]. Counterfactual studies alter the existence or value of contextual qualities or variables and examine how these changes impact the outcomes by comparing facts and counterfactuals [52]. If the fact is that the policy is not being implemented, the counterfactual is that the policy has been implemented for a length of time at the same moment [53]. Changsheng Xiong et al. confirmed the validity of counterfactual simulations by comparing the impact of an airport with and without construction on the surrounding land use [11]. Thus, this method was also applicable to this study. Based on the fact that Nanyuan Airport terminated its operations in September 2019, we generated the counterfactual that Nanyuan Airport did not cease operations in September 2019. The difference between the truth and the counterfactual is the impact of land use and cover on the surrounding area before and after the closure of Nanyuan Airport. Land use patterns over the same construction era could be simulated using a cellular automata (CA)-Markov model based on previous geographical and temporal variations in land use. As shown in Figure 4, impact (a) depicts the standard evaluation approach, which compares the land use pattern before and after the airport's decommissioning in 2019. In impact (b), we assume that the land use pattern will change in accordance with the inherent trend, and we simulate the land use pattern in 2022 using the actual land use pattern in 2011 and 2019 and compare it to the actual 2022 land use pattern to derive the difference in the impact on the surrounding land use pattern before and after the airport shut down. It is worth mentioning that Figure 4 only depicts general trends in land use patterns, and the changes are not necessarily linear. Moreover, changes in land use patterns may be impacted by other variables, such as specific land use rules or factors such as population and economic growth, which were not considered in this study and will need to be investigated in future work.

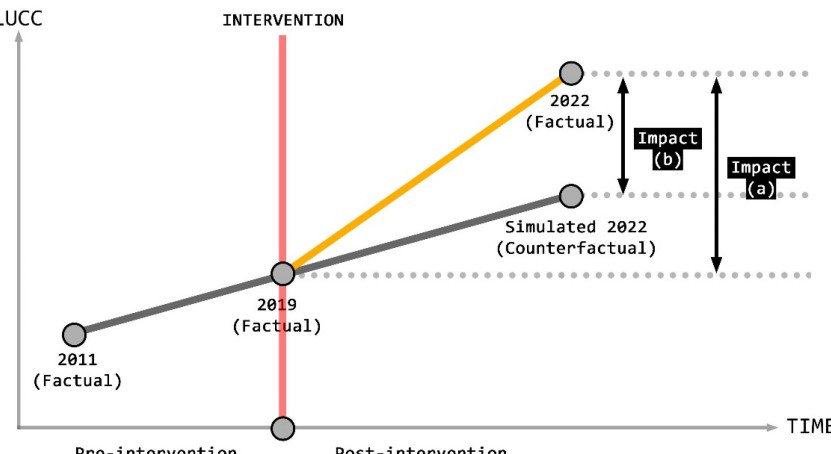

**Figure 4.** Changes in the airport's influence on the surrounding land use pattern were evaluated using conventional comparison (a) and counterfactual simulation (b).

*3.3. Research Framework*

The research methodology and steps are summarized in Figure 5.

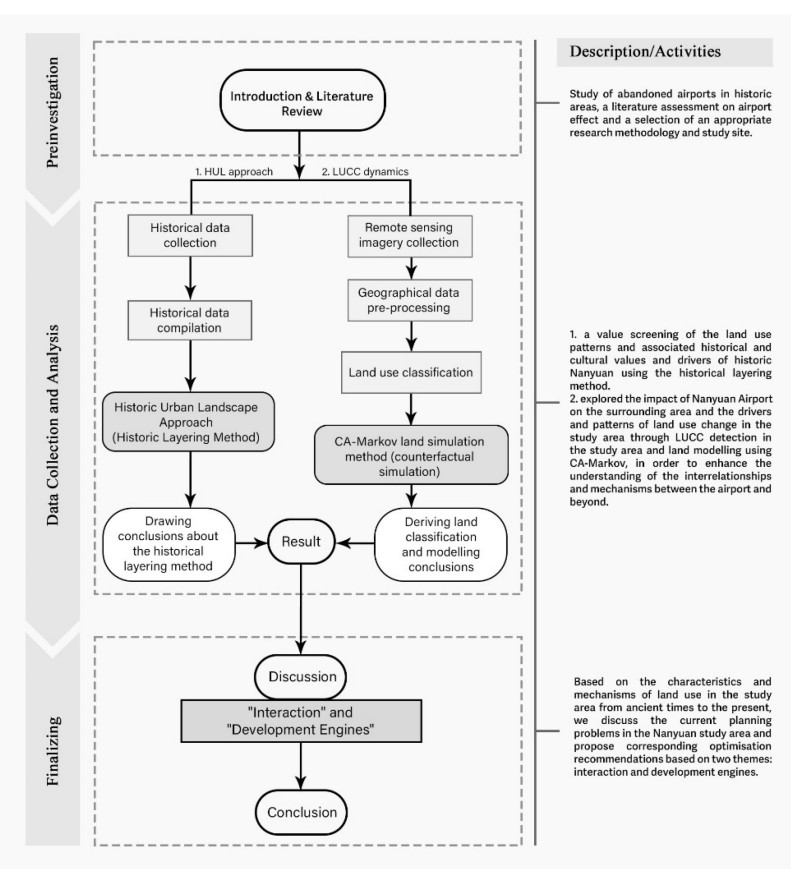

**Figure 5.** Research framework.

## 4. Result

*4.1. Evolution of Land Use Patterns and Development of Historic Urban Landscape Elements in the Nanyuan Airport Study Area*

The construction of Chinese capitals has always included geographically favorable areas in the overall layout, and these areas are interdependent with the capital city, forming an organic whole [54]. The Nanyuan Airport study area includes primarily the Royal Garden

of Nanyuan, the site of the five dynasties' royal hunting grounds and the imperial capital gardens of the Yuan, Ming, and Qing dynasties [54]. The extent of Nanyuan throughout history is shown in Figure 6.

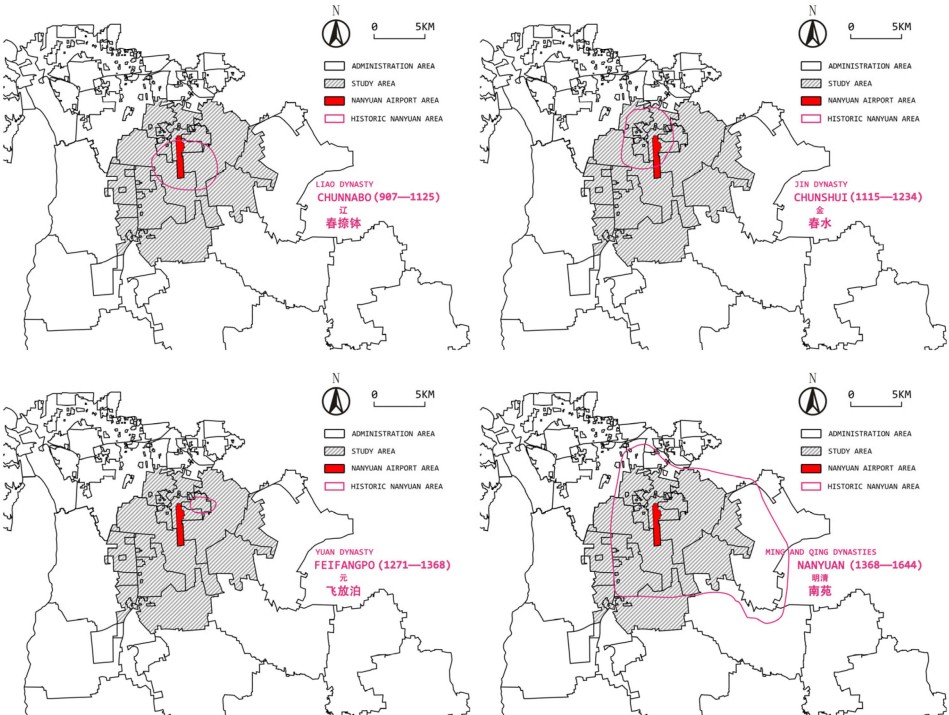

**Figure 6.** A diagram of the evolution of the extent of Nanyuan throughout history [55].

4.1.1. Budding Period: Following Nature (Liao and Jin Periods)

According to records, during the Liao dynasty, the imperial family initially undertook hunting and martial training in the Nanyuan region, located south of the city [55]. At the time, the oscillations of the old riverbeds, which were low-lying and littered with waterbanks over hundreds of kilometers, influenced the region around Nanyuan [56]. To retain their total supremacy in the military and to uphold their traditions, the emperors needed to build a hunting area and a location to practice martial arts near the palace, for which Nanyuan, on the capital's southern outskirts, was the best choice, which also led to the formation of the Nabo activities [57]. This also led to the development of the Nanyuan area and the creation of a distinctive culture of hunting.

The Jin dynasty's rulers continued to develop the land in the Liao dynasty's tradition, and the emperor's frequent visits here were destined to result in the construction of palaces in which to meet his ministers and deal with government issues. Hence, the Jin dynasty emperors erected the spring palace and some other palaces here, as the southern part of the palace, to continue fishing and hunting activities, called spring water and autumn mountain [58,59]. Since then, the relevance of hunting activities has been maintained throughout Nanyuan's development.

4.1.2. Period of Development: Delineation of Boundaries (Yuan Period)

During the Yuan dynasty, the site was known as 'Fei Fang Po' [60]. In addition to continuing the hunting traditions of the Liao and Jin dynasties, the Yuan emperors also built a small number of buildings. According to the Yuan Hunyifangyu Shenglan, by this time, Fei Fang Po already had clearly defined boundaries. The Yuan rulers began demarcating royal hunting grounds, prohibiting commoners from privately hunting within them [61], thus establishing the history of Nanyuan as an imperial park. However, due to the nomadic nature of the culture, there was no extensive gardening and the original natural landscape and wetlands remained largely intact [62,63].

### 4.1.3. Maturity: Gradually Perfected (Ming and Qing Dynasties)

The Nanyuan area was known as 'Nanhaizi' during the Ming dynasty, and the Ming dynasty extended the halls and official chambers at Fei Fang Po and erected a wall around it of approximately 70 km, enclosing Nanhaizi as an imperial garden [64]. As the remnants of the hostile Mongolian forces outside of the border still existed, while many aspects were still influenced by the culture of the northern minorities, the Ming dynasty's Nanhaizi still retained its hunting and military training functions. In addition to this, a productive function was added, with over 1000 people, called Haihu, recruited to protect the royal family's gardens, maintain animals in captivity, and harvest fruit and melons [62].

The Qing dynasty took official control of Nanyuan in the 13th year of Shunzhi's reign (1656). After the twenty-third year of the Kangxi reign, it progressively became a site for military training and parades, as well as the garden's logistical backbone. The Qing dynasty's rulers developed Nanyuan in accordance with the local conditions, and the area inside it was mainly utilized as a natural hunting environment, as seen in Figure 7. At the same time, part of the land in Nanyuan was also developed into cultivated land for the imperial estate, mainly for the use of the imperial palace [65], and part was left as wild pasture for livestock breeding, as shown in Figures 8 and 9. Furthermore, the Qing emperors were acutely aware that the Nanyuan water system was a critical link in the management of Beijing's water system, and they hence conducted a large-scale systematic upgrade of Nanyuan's water system [66]. The main functions of Nanyuan during the Qing dynasty are summarized in Figure 10.

In 1902, the Qing government ordered the establishment of the Nanyuan Bureau of Reclamation to lease the unused land in Nanyuan and to cultivate it so that the people could survive and make a living. Eunuchs, dignitaries, landlords, and businessmen from the palace flocked to the area, enclosing land and building dozens of feudal landlords' estates in Nanyuan in succession. As a result, many villages began to appear in Nanyuan. After the Jiaqing period, Nangyuan's hunting and martial arts functions were significantly reduced, and it mostly served as a military cantonment in the south of Beijing. When the Eight-Power Allied Forces invaded Beijing, they destroyed all the palaces and temples in Nanyuan [67]. In 1910, the Qing government set up an airstrip in Nanyuan for aviation. This was the end of Nanyuan's period as a royal garden, and, with the establishment of the Nanyuan Aviation School in Nanyuan by the Beiyang government of the Republic of China, its fate as a 'development' took on a new historical dimension [68]. At the end of the Qing dynasty, Nanyuan no longer had the appearance of an imperial garden, but, due to its position, it became a military barrier and a testing ground for modernization in Southern Beijing.

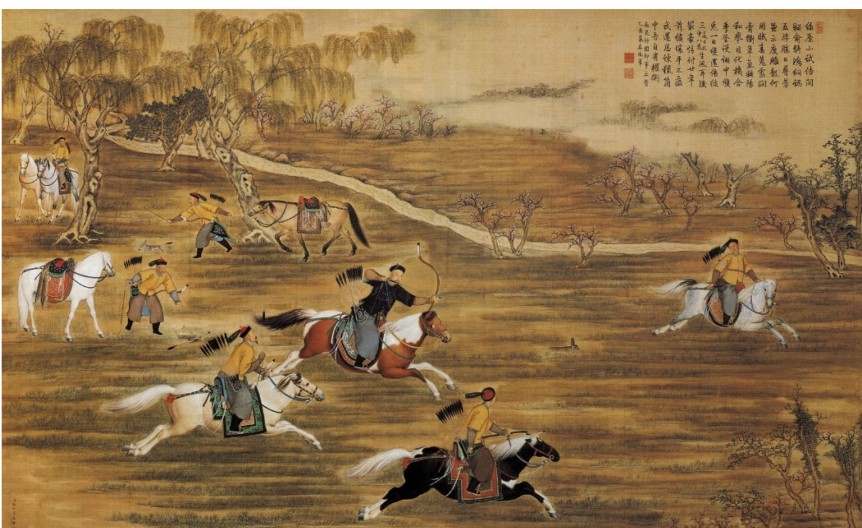

**Figure 7.** The Qianlong Emperor Hunting Hare by Giuseppe Castiglione (1755).

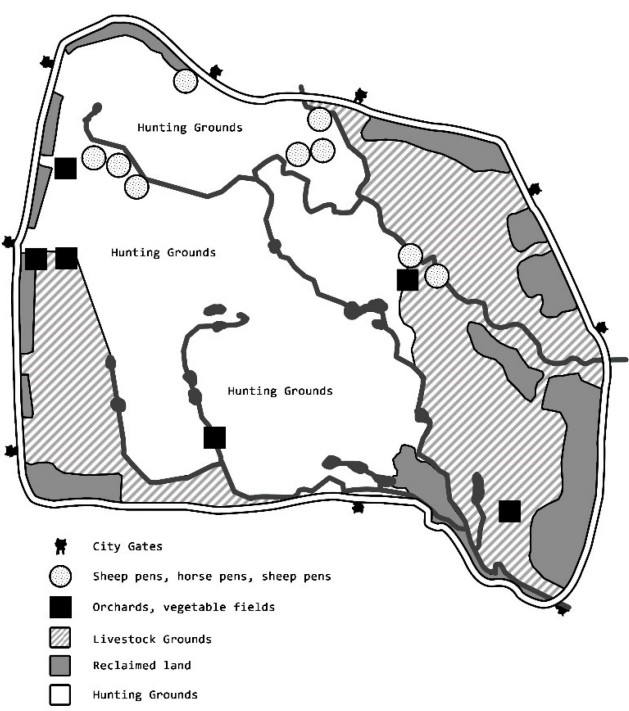

**Figure 8.** Zoning map of the Nanyuan site during the Kangxi period of the Qing dynasty (adapted from the First Historical Archives of China, "The Complete Map of Nanyuan in 1699").

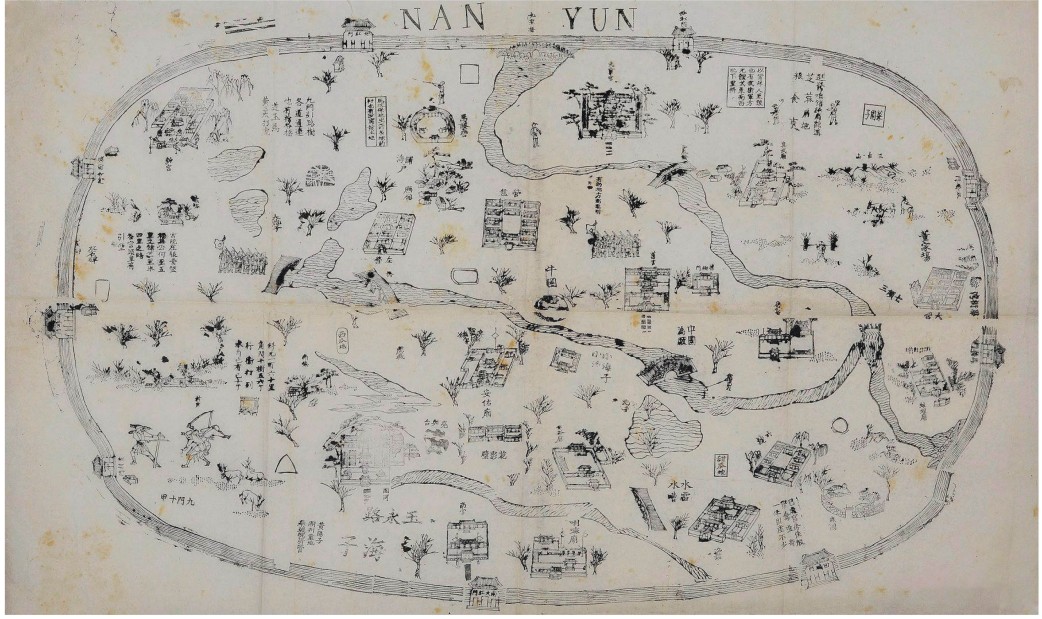

**Figure 9.** Map of the royal hunting grounds in Nanyuan, Beijing, painted on wood at the end of the Qing dynasty.

4.1.4. Decline: War-Torn (Modern Period)

In the early years of the Republic of China, the army occupied Nanyuan, continuing the function of training and parading troops from the Qing dynasty, converting it into barracks to train troops to guard the capital and also holding parades here on several occasions. At the same time, a railway station was built to strengthen the links between Nanyuan and Beijing and the rest of the country. Many train lines passed through Nanyuan, contributing to the economic, social, and cultural development of the area. In addition, Nanyuan was the birthplace of Chinese aviation. With the widespread use of aircraft in

the military, the Ministry of Transport of the Beiyang Government set up the "Office of Aviation Preparation" to develop civil aviation, making Nanyuan Airport not only the first military airport in Chinese history but also the first civilian airport, as shown in Figure 11. Subsequently, after the Lugou Bridge Incident in 1937, the Japanese expanded Nanyuan Airport and built Japanese barracks, small semi-underground bunkers, and more than 20 aircraft nests, transforming it into an important military base and a large airport for the invasion of Chinese territory [69]. Furthermore, Nanyuan has a distinct cluster of settlements based on the plain topography and abundant water resources, with names like 'Ying', 'Fa', and 'Zhuang' still being used in numerous places. The area with the word 'Ying' was once home to immigrants from Shanxi, who immigrated to the area dozens of times in the early Ming dynasty and set up battalions on the land for paramilitary management; the area with the word 'Fa' was an area of long-standing cultivation on the muddy land formed by the former course of the Yongding River in Nanyuan, which was turned over to create a fa; the area with the word 'Zhuang' was mostly a landowner's estate formed by the enclosure of land in the late Qing dynasty, as shown in Figure 11.

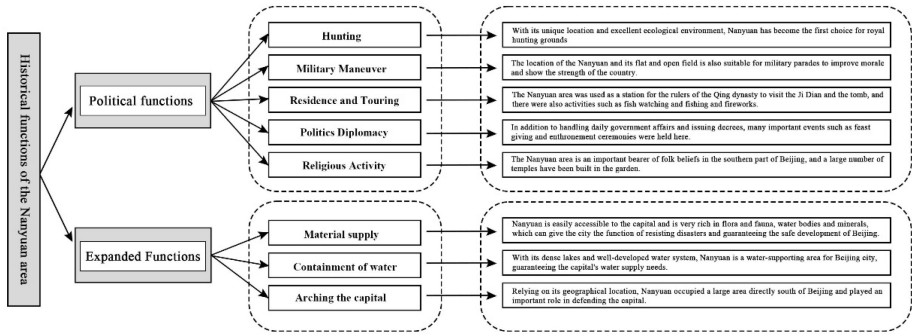

**Figure 10.** The main functions of Nanyuan during the Qing dynasty (self-drawn by the authors).

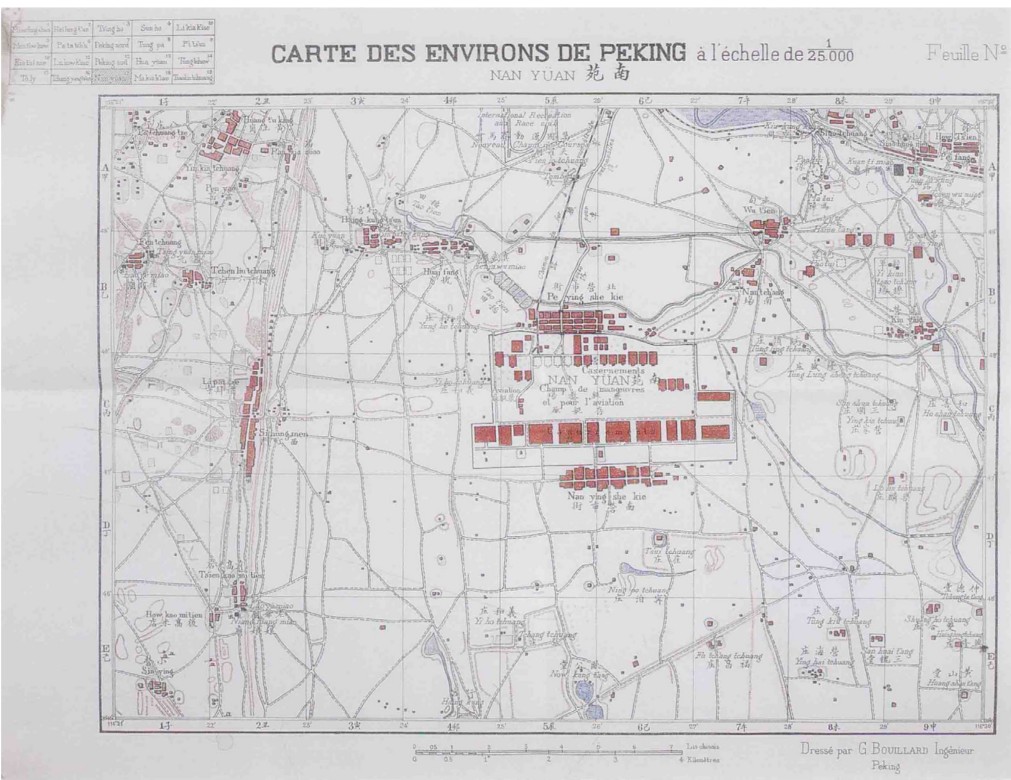

**Figure 11.** The map of Nanyuan (39 × 53 cm). Scale: 1:25,000 (By G.Bouillard Ingènieur, Beijing, 1923 in Carte Des Environs De Peking).

With the foundation of New China, Nanyuan Airport followed the historical tradition of establishing a flying school and an air force training facility and remained a military airport. The airport was not available for civil aviation until 1986, when China United Airlines was founded [70]. In 2011, the Nanyuan Airport renovation and expansion project began with the construction of a new terminal building. On 23 September 2019, the military part of Beijing Nanyuan Airport was officially transferred to Beijing Nanjiao Airport; on 25 September, Beijing Nanyuan Airport officially ceased to operate and was abandoned. The Nanyuan area no longer retains its original natural appearance, and the villages within the site have successively entered the suburbanization process, transforming into an urban area with a mixture of villages and towns, with a variety of different styles and features, and some of the villages are currently being demolished, while others are being rebuilt into communities.

4.1.5. Comparative Analysis of Land Use and Historic Urban Landscapes

Based on documentary evidence and archaeological findings, we stratified and analyzed the historical lineage of the area, overlaying it to form a complex and interrelated historical–spatial network of the Nanyuan area. In this study, we created a table that categorized the main historical and cultural landscape elements, as well as land use patterns, based on historical periods. The changes and continuity of these elements were indicated by color variations. This approach was employed to analyze the deep-seated correlations between the cultural geography, historical transformations, land use patterns, and other related factors within the research area of the Nanyuan Airport, as shown in Table 3. After conducting a comprehensive analysis of the various elements and data pertaining to the Nanyuan Airport study area, several distinct characteristics were observed in relation to its land use and management.

(1) The history of the development and evolution of the greater Nanyuan area is not a single fragment but encompasses long phases from the Liao, Jin, Yuan, Ming, Qing, and Republican periods to the present day, with a cultural fault emerging today. Through the construction of hunting grounds and parade grounds during the Liao and Jin dynasties, a solid foundation was laid for the development of the Nanyuan region, and the embryonic form of the historical urban landscape of Nanyuan emerged. With the delineation of the Nanyuan area during the Yuan dynasty, the historical urban landscape of Nanyuan took shape. Through the construction of the imperial hunting grounds during the Ming and Qing dynasties, as well as the decline of the Qing dynasty and the anti-aggression wars, the ancient historical urban landscape of Nanyuan matured, declined, and eventually came to an end, with the basic pattern of the ancient historical landscape of Nanyuan no longer existing. Finally, the Nanyuan region became a military stronghold on the modern outskirts of Beijing, an experimental site for railway and transportation, and the location of China's first military and civilian airport, leading to the development of the modern cultural landscape of Nanyuan.

(2) The cultural and geographical characteristics of the Nanyuan area have allowed the region to retain the essential characteristics of the "garden" in terms of land use patterns in ancient China. Unlike other large royal gardens, the management of Nanyuan by successive rulers was slow to develop and the scale of construction was minimal, with a maximum building density of around 0.027% [54]. Even when the treasury was full at the time of the Qianlong event, only two new buildings were built. This is a direct reflection of the non-architectural nature of the construction of Nanyuan, which was rooted in the nomadic culture of hunting and the tradition of martial discipline. Furthermore, because of the Nanyuan area's natural wetland environment and its privileged location south of the Forbidden City, Nanyuan evolved throughout the Liao and Qing dynasties, retaining almost all of the ancient functions of a garden, such as hunting and military parades, material production, and resource

supply reserves, and it has remained close to the original meaning of 'garden' in ancient China.

**Table 3.** Evolving land use patterns and the historic townscape in Nanyuan (The changes and continuity of the historical and cultural landscape elements as well as land use patterns were indicated by color variations).

| | Liao Dynasty (907–1125) | Jin Dynasty (1115–1234) | Yuan Dynasty (1271–1368) | Ming Dynasty (1368–1644) | Qing Dynasty (1636–1912) | Modern Times (1912–2023) |
|---|---|---|---|---|---|---|
| **Environment** | Natural wetland with abundant water and grass | Natural wetland with abundant water and grass Few buildings constructed | Natural wetland with abundant water and grass Few buildings constructed | Natural wetland with abundant water and grass Few buildings constructed | Natural wetland with abundant water and grass Scattered villages Water system improvement | A mix of village, town, and township |
| **History** | Chunnabo Activity | Chunshui Activity | Feifang Activity | Opening of Shanglingyuan and expansion of Nanhaizi | Development of Nanhaizi Establishment of Imperial Villa | The palace and temples were destroyed Build railway station and airport |
| **Culture** | Hunting Culture | Hunting Culture | Horsemanship and Archery Culture | Farming Culture | Horsemanship and Archery Culture | Confucian Culture |
| **Land Use** | Hunting Military Parade and Training Rituals | Hunting Military Parade and Training Rituals Garden tour | Hunting Military Parade and Training Rituals Garden tour | Hunting Military Parade and Training Rituals Garden tour Supply of goods Guarding the capital | Hunting Military Parade and Training Rituals Garden tour Political Diplomacy Water Conservation Supply of goods Guarding the capital | South Beijing Military Barrier Nanyuan Airport Forgotten urban area South Central Axis Extension |

(3) Land use and management in Nanyuan was based on an ecological foundation and focused on holistic thinking. The Ming and Qing capitals were not confined to the old city but were built in a wider area to construct a fully functional capital system. As a wetland, the management goals in Nanyuang were mainly based on ecological foundations and were adapted to the local conditions, with hunting, martial arts, and gardening as the mainstay, supplemented by material supply and ecological nourishment, so that it gave full play to the urban and ecological functions as an important part of the capital's functional system. As a result, Nanyuan has developed over a long period of time, interacting with the production and social structure of the capital, forming a unique spatial pattern and cultural system, and maintaining a high degree of integrity and complementarity with the functions and space of the capital.

(4) There is a correlation between the land management of Nanyuan and the landscape character of the area. The landscape and economic benefits were controlled by adjust-

ing the land use types and crop cultivation. Nanyuan was largely self-sufficient in its financial policies, except for the additional funds that it had to draw on from the Ministry of Finance [55]. However, the land within Nanyuan was not developed in an uncontrolled manner to meet financial needs, and, each year, the number of acres in Nanyuan is re-evaluated and adjusted. There are always acres that are continually reclaimed, as well as ripe land that is abandoned and converted into hunting grounds, ensuring a certain level of productive activity while the area's landscape is maintained, and the land use is always well proportioned and coordinated.

### *4.2. Land Use and Land Cover Change Dynamics*

### 4.2.1. Classification of Land Use and Accuracy Assessment of Land Modeling

As shown in Table 4, we initially evaluated the accuracy of land use categorization for 1986, 1993, 2001, 2011, 2019, and 2022. The overall classification accuracy was found to be 88.83%, 87.8%, 87.7%, 88.01%, 89.19%, and 86.83%, with Kappa coefficients of 0.86, 0.85, 0.86, 0.87, 0.87, and 0.84. Consequently, the accuracy of this study's land use categorization satisfied the criterion of at least 80% accuracy of sensor data [71], and it was appropriate for further investigation.

In addition, for the accuracy assessment of the land use pattern simulated by the CA-Markov model, we compared the area of the real and simulated land use and land cover classifications for 2019, as shown in Figure 12 and Table 5. Comparing the results of both showed that the rate of change in area for each land use category was less than 10%. Simultaneously, we utilized IDRISI Selva's VALIDATE module to analyze the simulation's correctness, and the findings revealed that the overall K value between the two was 0.8162 at the 1% significance level, which was higher than 0.75, indicating that the simulation's accuracy was acceptable [72,73]. As a result, we concluded that the CA-Markov model utilized in this investigation was applicable to this research.

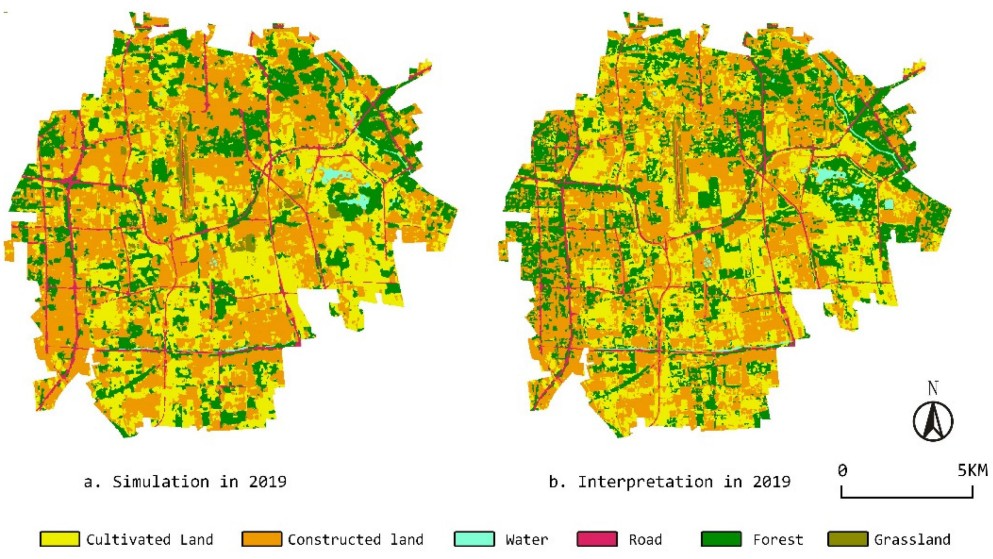

**Figure 12.** The 2019 land simulation and realistic land use classification results.

**Table 4.** Accuracy assessment for 1986, 1993, 2001, 2011, 2019, and 2022 classified images.

| Land Use and Land Cover | 1984 | | 1993 | | 2001 | | 2011 | | 2019 | | 2022 | |
|---|---|---|---|---|---|---|---|---|---|---|---|---|
| | User Accuracy | Producer Accuracy | User Accuracy | Producer Accuracy | User Accuracy | Producer Accuracy | User Accuracy | Producer Accuracy | User Accuracy | Producer Accuracy | User Accuracy | Producer Accuracy |
| Cultivated land | 92.9 | 90.6 | 92 | 88.1 | 93.4 | 83.9 | 89.9 | 92.7 | 86.7 | 83.4 | 89.2 | 83.1 |
| Forest | 92.8 | 84.4 | 84.9 | 88 | 83.3 | 89 | 82.3 | 92.9 | 90.8 | 88.7 | 82.6 | 88.8 |
| Grassland | 85.1 | 90.5 | 88.2 | 84.1 | 88 | 83.2 | 89.4 | 90.9 | 88.1 | 91.4 | 90 | 88.7 |
| Road | 83.9 | 84.6 | 85.6 | 82 | 87.8 | 82.8 | 83 | 90 | 88.8 | 90.1 | 86.2 | 82.8 |
| Constructed land | 89.7 | 87.9 | 84.9 | 93.1 | 93.8 | 87.2 | 83.4 | 90.1 | 85.2 | 92.8 | 87.7 | 94 |
| Water | 91.9 | 91.7 | 92.6 | 90.1 | 86.9 | 93.2 | 87.9 | 83.4 | 91.9 | 92.4 | 83.1 | 85.8 |
| Overall accuracy (%) | 88.83 | / | 87.8 | / | 87.7 | / | 88.01 | / | 89.19 | / | 86.83 | / |
| Kappa coefficient | 0.86 | / | 0.85 | / | 0.86 | / | 0.87 | / | 0.87 | / | 0.84 | / |

### 4.2.2. Land Use and Land Cover Classification and Simulation Results

The results of our land use and land cover classification for the study area for the years 1984, 1993, 2001, 2011, 2019, and 2022 are shown in Figure 13, revealing that the most significant changes in land use patterns by year are for constructed and cultivated land. Since 1984, new construction land has increased mainly on the north and southwest sides of the study area; however, by 2001, the development of construction land on the north and southwest sides of the study area had reached saturation and growth had slowed, while construction land on the east and south sides had gradually increased. At the same time, cultivated land loss was concentrated in these areas. Furthermore, the area of forest land is also highly variable, showing a year-on-year increase. Figure 14 depicts the change in area for each of the six site types by study year, demonstrating in further detail that the amount of constructed land in the research region has been growing year after year, peaking in 2001 and then dropping, but returning in 2022. The change curve for cultivated land, on the other hand, is the inverse of that for constructed land, beginning with a drop, followed by a steady rebound in 2001, and then declining again in 2022. The curves for grassland, forest, and road areas are smoother: the growth rate of grassland areas gradually flattens out, while the growth rate of forest land areas slows in 1993, picks up in 2001, and then slows again in 2019; the growth rate of road areas increases each year and then gradually slows after 2011.

**Table 5.** Comparison of simulation and interpretation data in 2019 (area in ha).

| 2019 | Cultivated Land | Forest | Grassland | Road | Constructed Land | Water | Total |
|---|---|---|---|---|---|---|---|
| Interpretation | 5345.46 | 4522.59 | 99.765 | 608.0175 | 6984.20 | 159.525 | 17,719.56 |
| % | 30.17% | 25.52% | 0.56% | 3.43% | 39.42% | 0.90% | 100% |
| Simulation | 5008.898 | 4091.575 | 149.065 | 860.2425 | 7374.32 | 135.46 | 17,719.56 |
| % | 28.27% | 23.09% | 0.84% | 4.86% | 41.16% | 0.76% | 100% |
| Area change | 336.5625 | 431.015 | −49.3 | −252.225 | −390.1175 | 24.065 | 0 |

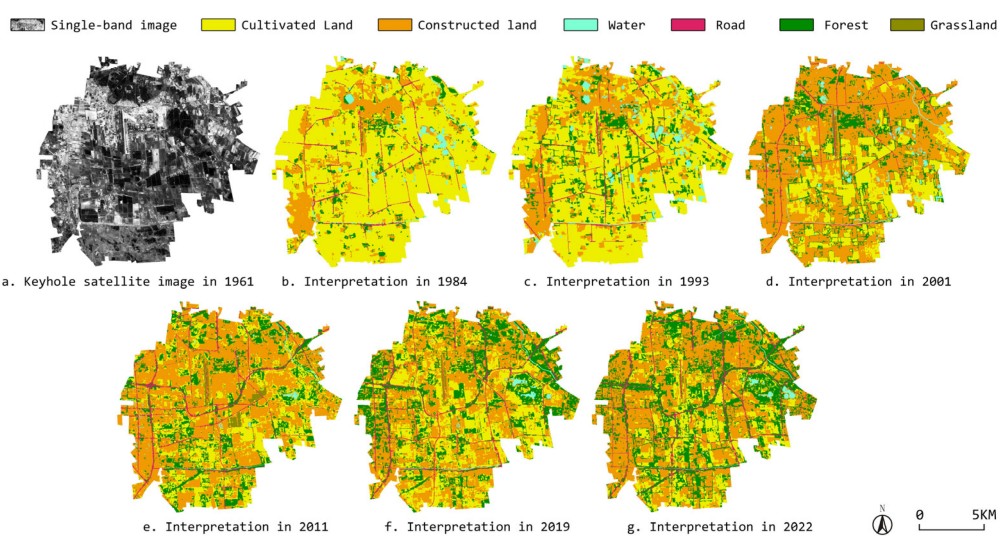

**Figure 13.** (**a**) Keyhole satellite images and land use and land cover maps based on (**b**) 1984 TM images; (**c**) 1993 TM images; (**d**) 2001 ETM images; (**e**) 2011 ETM images; (**f**) 2019 OLI_TIRS images; (**g**) 2022 OLI_TIRS images in Nanyuan Airport study area.

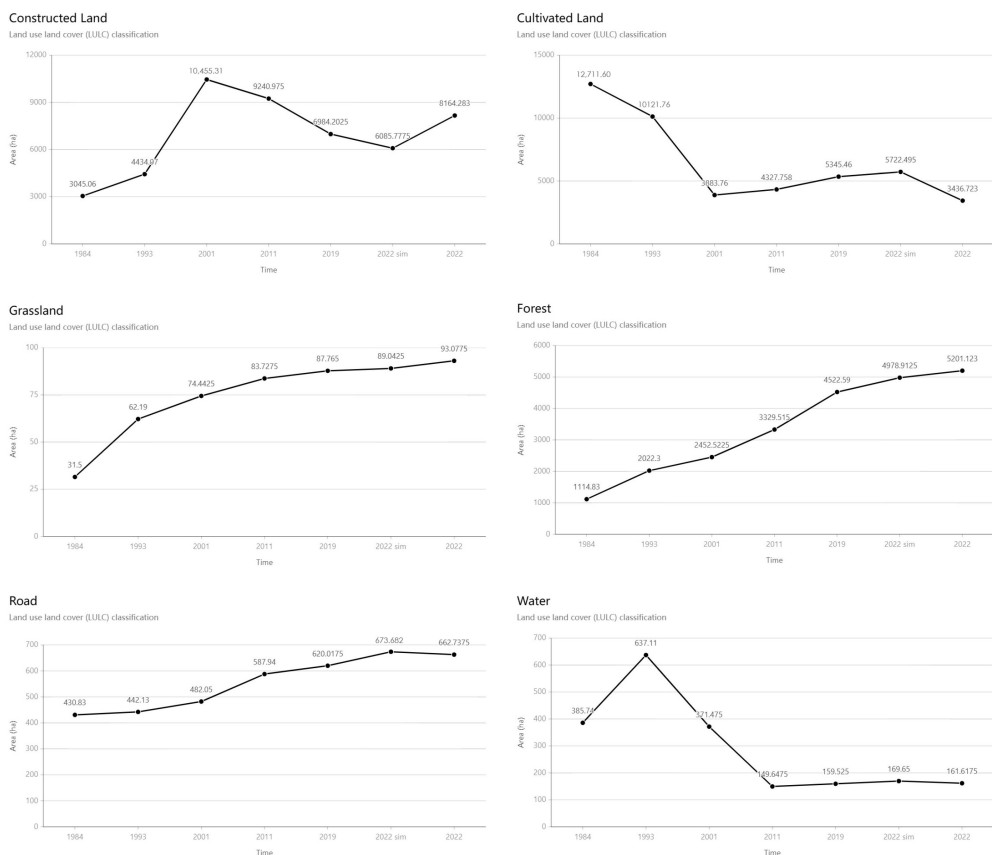

**Figure 14.** Real and simulated changes of various land use and land cover classifications for 1984, 1993, 2001, 2011, 2019, and 2022.

We then used a Markov model to calculate the corresponding land use transfer matrix based on the land use data for each study year, and we converted it into dependency wheels, as shown in Figure 15, which shows the changes in land use flows within the study area between different years. For example, between 1984 and 1993, the primary land use category in the study region was cultivated land. Some constructed and forest land was converted into cultivated land, while some cultivated land was changed to forest land and constructed land. Between 1993 and 2001, a notable trend of land use conversion from constructed to cultivated land was observed in the study region. Over time, the proportion of constructed land steadily increased, eventually surpassing that of cultivated land. Between 2001 and 2011, and between 2011 and 2019, the quantity of constructed land in the research area predominated, with a large amount of cultivated land replaced by new construction land and a substantial amount of construction land recovered as cultivated land. Moreover, there was a significant degree of interconversion between forest land, construction land, and cultivated land. After the decommissioning of Nanyuan Airport in 2019–2022, although the time span in Figure 15e is only three years and the proportion of land conversion is relatively small, similar land conversions continue to occur, which are likely to be due to factors other than the airport's impact, such as population and economic growth, and other land use policies or government-related decisions [11].

As illustrated in Figure 16, we utilized the CA-Markov model to predict the land use pattern in 2022 using the land use patterns in 2011 and 2019. By comparing their percentage differences in area, as shown in Table 6, we discovered that the difference between cultivated land and constructed land was greater than 10%, implying that if the study area is home to an airport in the future, the area of cultivated land will increase, while the area of constructed land will decrease. The truth, however, is quite the contrary. Moreover, the overall land use pattern indicates that constructed land remains the dominating land use category in the study region, with cultivated land being more fragmented in reality than

in simulation. The other land use groups do not differ significantly, with all being less than 2%. Hence, from a counterfactual standpoint, the airport's decommissioning has largely loosened the limits on the expansion of constructed land, resulting in a reduction in the cultivated land area. At the same time, by comparing the traditional approach with the counterfactual approach, as shown in Figure 17, one can see the inconsistency in the direction of impact of constructed and cultivated land and other categories. The traditional approach indicates a decline in cultivated land and a rise in constructed land, while, in the counterfactual evaluation approach, cultivated land seems to be retained more, and constructed land instead decreases. On the other hand, the road and water land categories seem to develop more rapidly.

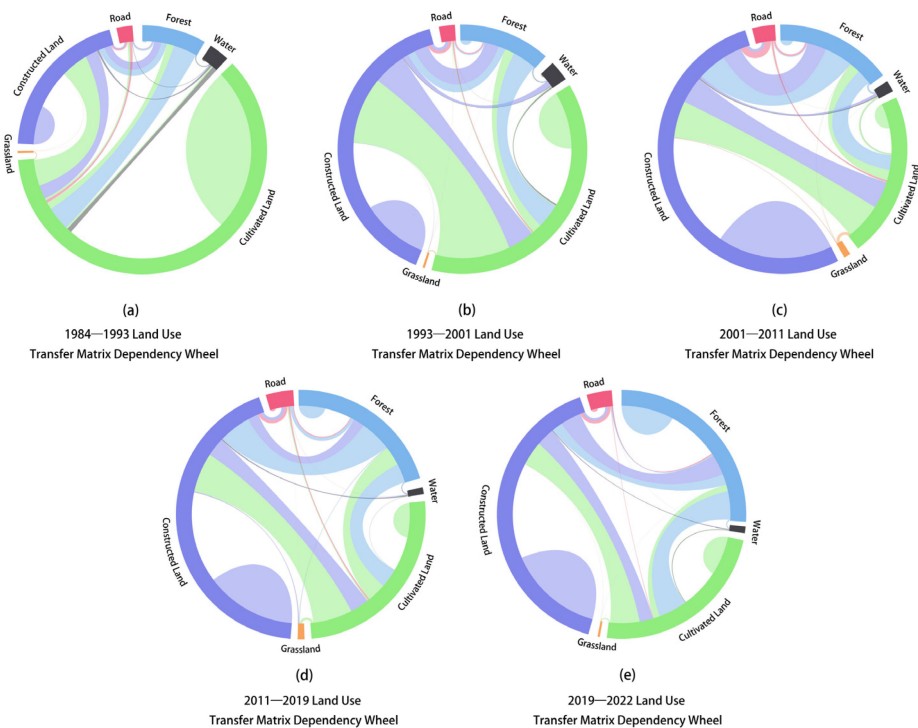

**Figure 15.** Dependency wheels for land use flow changes among different years in the study area.

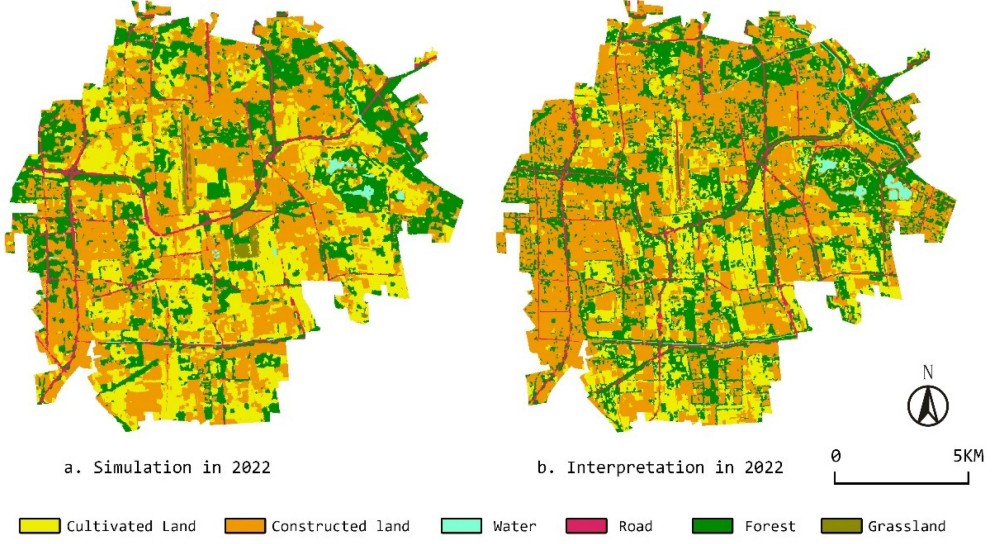

**Figure 16.** The 2022 land simulation and realistic land use classification results.

**Table 6.** Comparison of simulated and real land use area and percentages in 2022 (Note: area variance is simulated 2022 land use land cover (LULC) area minus real 2022 LULC area; percentage variance is simulated 2022 LULC minus real 2022 LULC area percentage).

| Land Use Type | 2022 Sim LULC | | 2022 LULC | | Area Variance | Percentage Variance |
|---|---|---|---|---|---|---|
| | Area | % | Area | % | | |
| Forest | 4978.9125 | 28.098% | 5201.12 | 29.352% | −222.21 | −1.254% |
| Water | 169.65 | 0.957% | 161.6175 | 0.912% | 8.03 | 0.045% |
| Cultivated Land | 5722.495 | 32.295% | 3436.72 | 19.395% | 2285.77 | 12.900% |
| Grassland | 89.0425 | 0.503% | 93.0775 | 0.525% | −4.04 | −0.023% |
| Constructed Land | 6085.7775 | 34.345% | 8164.28 | 46.075% | −2078.51 | −11.730% |
| Road | 673.682 | 3.802% | 662.7375 | 3.740% | 10.94 | 0.062% |
| Total | 17,719.56 | 100.000% | 17,719.56 | 100.000% | 0.00 | 0.000% |

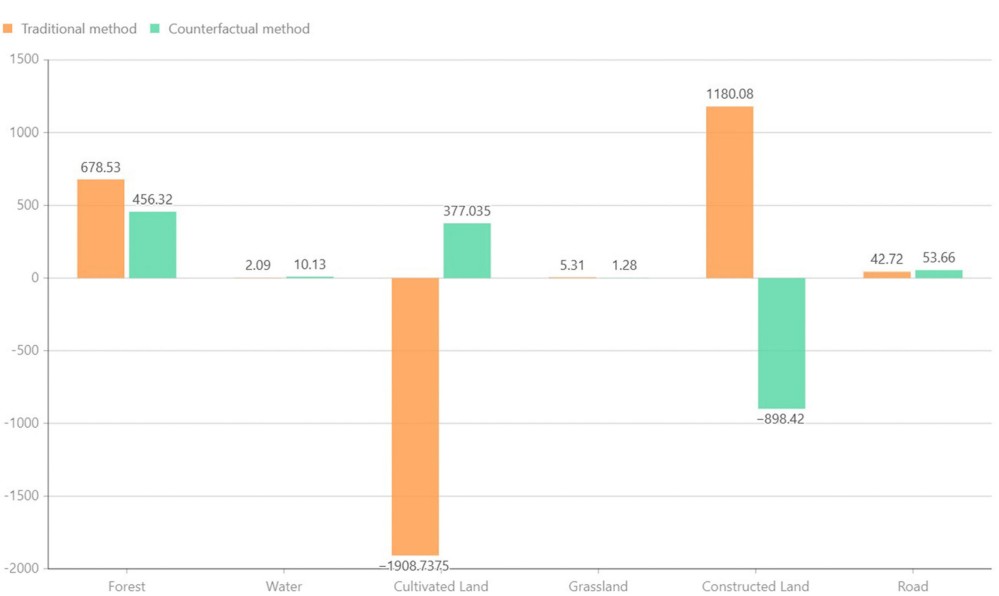

**Figure 17.** Land use change analysis under different methods; the traditional method compares real land use patterns in 2019 and 2022, and the counterfactual method compares real 2019 and simulated 2022 land use patterns. Note: Data shown in the figure are simulated or real 2022 LULC land use data minus real 2019 LULC land use data in hectares.

4.2.3. Interrelationships, Mechanistic Features, and Causes within and outside the Airport

In summary, we can identify various aspects and causes for the interrelationships and mechanisms within and beyond the airport in the Nanyuan study area based on the results presented above.

(1)    Over its long history, Nanyuan Airport has had a profound influence on the basic pattern and landscape character of the site. Throughout the research period, the frequent conversion of cultivated land to constructed land was a major feature of the land use change during the study period. Since the construction of the airport, the land use of the airport itself and its surrounding areas has been influenced by territorial effects and aviation externalities, resulting in disruptions to ground transportation networks, impediments to urban development, spatial distortions in the urban structure around the airport, and the formation of noise landscapes characterized by agriculture, infrastructure, or low-density areas within the noise contour lines [2]. Furthermore, the



study region has been impacted by population and economic expansion, as well as other land use policies and decisions. In the survey, we can see that the study region has witnessed drastic land use changes since 1984, as illustrated in Figure 13. This transformation can also be found to have persisted to some extent to the current day, with the related environmental and cultural influences leaving an indelible mark on the area.

(2)  The construction and development of Nanyuan Airport, as well as the degree of urbanization in the surrounding area, had a one-of-a-kind influence on the land use patterns. According to the data, the degree of urbanization around the airport was relatively low between 1984 and 2001, and the airport had recently been converted from a military to a civilian–military airport, with a reduced capacity; thus, it had less of an urbanizing effect on the surrounding area, as shown in Figures 13 and 14, which depict a typical urbanization pattern. However, during the period 2001–2011, Nanyuan Airport experienced the resumption of flights and the renovation and expansion of the terminal building, which greatly increased the number of flights carried and the scope of the airport's influence on the surrounding area increased. The economic or traffic locations around the airport may have been rebuilt as a result. At the same time, the urbanization of the surrounding area reached a new stage, resulting in a reduction in the amount of land available for construction during this period and an increase in the amount of cultivated land. After the airport's shutdown in 2019, many urban development restrictions were lifted, and, in only three years, the amount of constructed land grew significantly, while cultivated land declined rapidly. Although we cannot quantify how much of the fluctuation in the amount of different land uses is due to physical or urban effects from the airport's closure versus other land policies, the land simulation results definitively demonstrate the influences imposed by the airport on its surroundings. It is clear that urbanization and population growth, as well as the strong demand for new housing and the physical position of the study area in Beijing's south central axis, have increased the rate of land use reconversion to new urban development and urban parks. The airport's shutdown has created a new opportunity for local development.

(3)  China's specific land policies have led to particularities in the changing land use patterns in and around Beijing's Nanyuan Airport. The substantial increase in forest land use within the research area is attributed to various policies implemented by the Chinese government, ranging from the 1981 Resolution on Compulsory Tree Planting for the Whole Country to the National Forest City Construction Policy [74]. These policies have facilitated the expansion of forest land through extensive land clearance and reclamation efforts. At the same time, the government's absolute power in construction has also been applied to the airport and its surroundings. For example, the expropriation of farmland near airports involves China's land policy on balancing the occupation and compensation of cultivated land (gengdi zhanbu pingheng, CLRCQ) [75], which is the process of supplying an equivalent amount and quality of cultivated land to local governments for the purpose of development. There is also a land acquisition system (tudi Zhengshou, LE) [76], in which local governments acquire land from farmers and residents for airport construction and then compensate them in a certain amount. On the one hand, the overall change in the airport's construction is government-led, and, therefore, the maintenance of land development and land acquisition surrounding the airport is largely restricted by the government's budgetary conditions. However, under a government-led land use mode, it is simpler to plan for land purchases, decommissioning, and future developments to the land around the airport.

## 5. Discussion

Urban development is a continual process of succession, and the historical legacy has developmental importance as well [77]. Regarding the planning and redevelopment

of Nanyuan Airport and its surrounding areas after its closure, it is crucial not only to assess the current conditions of the airport itself and the surrounding land use, but also to consider Nanyuan's historical status as an imperial hunting ground and royal garden over multiple dynasties, as well as its cultural attributes. Based on the results of both methods, we have obtained two sets of suggestions for the planning and redevelopment strategies of the Nanyuan study area: connections and developmental engines.

### 5.1. Connections

Based on the HUL investigation and land classification and simulation, we can clearly see that the environment within the Nanyuan study area has transformed from the "lush and verdant imperial hunting grounds and gardens" to rural villages deeply impacted by the airport, with the overall spatial layout undergoing fluctuations and changes over time. However, with the vigorous economic development and disordered introduction of modernist styles, many urban areas within the study area have lost the spatial awareness of the traditional Nanyuan, which is not conducive to sustainable regional development. After a comprehensive evaluation, we attempt to propose a framework to reconnect the ancient and modern landscape patterns of the study area, as well as the "connections" between different zones and functions, including three aspects.

(1)   Restore and strengthen the organic connection between the Nanyuan study area and the Beijing city proper. The Nanyuan study area should be restored as an important ecological functional zone in Southern Beijing. Through the historical stratification analysis, we can see that the Nanyuan area, as a direct extension of Beijing's functions, has had the purpose of ecological protection since its inception. In the hundreds of years of construction that followed, it maintained the minimum level of architectural construction and resource extraction. The Nanyuan wetlands once greatly replenished Beijing's groundwater [78], but, due to the cut-off of the Yongding River and land reclamation, the wetlands no longer exist, impacting Beijing's groundwater levels. At the same time, Beijing's urban spatial pattern of "northern mountains and southern waters, northern culture and southern martial arts" carries the city's own cultural memory and historical heritage, and restoring the overall wetland ecological pattern will benefit the continuation and development of Beijing's future urban context. On the other hand, through land use classification and simulation, we found that in only three years, from 2019 when Nanyuan Airport ceased operations to 2022, construction land increased rapidly with the high land use demand, especially in the west, north, and northeast of Nanyuan Airport, with the total area expanding from 6984.20 hectares to 8164.28 hectares (+16.90%). According to the counterfactual simulation results, the actual distribution of agricultural land is more fragmented. Although our research shows that the airport's shutdown has provided the locality with a brand new development opportunity, the primary focus of current planning should be on constructing a diversified spatial planning system, optimizing the land use structure, and scientifically and rationally resolving the contradiction between the growth of construction land and the protection of agricultural land. Considering the protection of the existing urban fabric, the appropriate promotion of the decentralization of non-capital functions and population control, the implementation of spatial integration and ecological restoration, and the focused shaping of existing ecological landscape areas such as Nanyuan Forest Park, as the core area of Nanyuan's landscape, are necessary to build an urban ecological network of "ecological zones–ecological corridors–ecological nodes".

(2)   Reinforce the linkages between Nanyuan's aviation cultural landscape, ancient Nanyuan's cultural landscape, and other historical urban landscapes. Our HUL analysis found that Nanyuan maintained the fundamental features of the garden landscapes seen throughout ancient Chinese history, albeit with extremely gradual construction. Meanwhile, our LUCC analysis revealed monumental overall pattern changes since 1961 as the dominant land use shifted from agricultural to construction land, with

Nanyuan Airport exerting a major influence on local land use models, as shown in Figure 13. These combined methods indicate that the research area currently suffers from severe cultural discontinuity. From the HUL perspective, culture-based urban development patterns are crucial for sustainable cities in the post-industrial era. As the epitome of the ancient Chinese central axes, the central axis at which the research area lies constitutes an important component of the city's cultural and historical heritage. Some scholars have pointed out that, since modern times, natural advantages evolved into human-driven forces as the foremost influences on central axis development and conservation in Beijing, often neglecting natural elements like overall landscape patterns. As an integral part of the southern central axis, the research area should respond to the extant aviation culture by reviving the ancient Nanyuan landscape and royal garden culture from five dynasties ago, along with traditional historical folk culture. Hence, during replanning, overall continuity and unity should be considered, unifying the "history", "present", and "future" within the Nanyuan context. Microscopically, methods like linking and weaving could incorporate historical zones, paths, and nodes with local production, society, economy, and nature into an integrated network. Additionally, the hierarchical relationships between cultural landscape developments across different levels and dimensions warrant further research and refinement.

*5.2. Development Engines*

The Nanyuan research area is situated on Beijing's southern central axis in the capital's urban layout, forming a vital spatial corridor linking the core districts, municipal subcenters, the new Daxing International Airport, and the Xiong'an New Area. The closure of Nanyuan Airport poses new challenges and opportunities for this zone. We endeavor to offer suggestions for the construction of a novel impetus to transform and develop the Nanyuan research area from tangible and intangible perspectives.

(1) Fully harness the physical value of the research area as a tangible engine for development. The innate physical attributes of Nanyuan Airport itself, including its scale, leftover infrastructure, and advantageous geographical location, are invaluable for the transformation and renewal of the Nanyuan urban zone. Our analysis of the land use surrounding Nanyuan Airport shows that the research area's environs are undergoing rapid urbanization, with a sizable population close to the city center, resulting in favorable development conditions that could support the redevelopment of the airport site. Notably, our findings indicate that China's unique land policies, such as the cultivated land occupancy compensation balance policy and land requisition system, also provide support for functional upgrades in the research area, albeit with some constraints, since maintaining land development around the airport and expropriation will be largely limited by governmental fiscal conditions. Moreover, based on our exploration of Nanyuan's historical stratum, we find that the research area's ecological ethos stemming from ancient Chinese garden appreciation awaits revival, in line with modern ecological leisure concepts. Hence, rather than rebuilding the airport into a new residential form or regarding it as an ordinary abandoned site by completely erasing its physical features, we believe that the Nanyuan Airport ruins should be considered as latent reserves for green and public spaces on Beijing's southern central axis. Specifically, adaptive reuse could transform it into a "cultural creative industry park and civic ecological park", making it an important leisure, recreation, and creative exhibition space for Beijing's southern region. Meanwhile, the areas surrounding Nanyuan Airport could also capitalize on their advantageous geographical location, taking the southern central axis as the guiding principle and blue–green ecological spaces as the foundation. Making full use of existing airport leftovers like rail transport and road infrastructure to construct high-quality urban service areas would serve to narrow the development gaps between Beijing's northern

and southern zones to achieve sustainable growth, fulfilling the site's developmental attributes while controlling fiscal expenditures.

(2) Fully harness the research area's inherent cultural value as an intangible engine for development. Unlike other airports, Nanyuan Airport has amassed diverse value and uniqueness through its centuries of evolution, with different eras of usage, aesthetics, and technical history. Likewise, the Nanyuan research area has undergone geographical landscape transformations, from royal gardens abundant in rivers, lakes, and springs to expansive farmland and clustered villages, which would also precipitate corresponding shifts in place identity. The sense of place emerging from human–space interactions and the associated emotional attachments stem fundamentally from the distinctive relationships between people and settings following cultural and social metamorphoses [19,79]. Thus, strong local identity facilitates regional sustainable development and forms associated local characteristics, propelling the area's further advancement. Consequently, the Nanyuan research area's urban development necessitates addressing the issue of urban homogenization, requiring the Daxing and Fengtai district governments to collaborate in constructing an integrated local cognition as the basis for new urban planning and development, thereby safeguarding the region's historical continuity and ensuring that residents have accurate perceptions of Nanyuan's distinct traits.

## 6. Conclusions

In this research, we utilize Beijing's Nanyuan Airport as a case study for the planning and regeneration of an abandoned airport area and its surroundings, as it is located in a historically and culturally rich region. We examine the applicability and respective advantages and disadvantages of the historic urban landscape (HUL) and land use and land cover change (LUCC) with counterfactual simulation approaches, and we attempt to combine the two approaches in order to deepen our understanding of the ancient and modern land use patterns in and around the airport.

Using the historical layering method, we were able to identify the features of the ancient Nanyuan area's land use patterns. Throughout the Liao and Qing dynasties, Nanyuan was an organic element of Beijing, with land management based on ecological foundations and nonbuilding construction as the dominating thread, while simultaneously retaining a high degree of integrity and complementarity with the capital's functions and spaces. Moreover, regional planning considers the balance between land development and the preservation of the area's landscape character. In modern times, however, the historical pattern of the land in ancient Nanyuan was lost due to political, economic, social, and cultural changes. In its place, Nanyuan Airport and the urbanization of the area have influenced the land, and the environmental history of Nanyuan has taken a significant turn, disturbing the original balance. Through comparative and counterfactual simulation methods of land use and land cover change, it was found that Nanyuan Airport had a profound impact on the basic pattern and landscape character of the site. As a whole, the frequent interconversion of cultivated land and constructed land was a prominent aspect of the land use change throughout the research period. In addition, the construction status of the airport itself and the degree of urbanization of the surrounding area had a unique impact on the land use patterns in the surrounding area, especially after the airport's decommissioning lifted the restrictions in some areas. At the same time, we also found that a large part of the specificity of the land use pattern in and around Beijing's Nanyuan Airport is due to the land policy on balancing the occupation and compensation of cultivated land, but also the land acquisition system in China. Based on the strategic goal of the sustainable development of the study area, we propose that the development of the area requires a functional orientation that exploits and inherits the historical and cultural lineage of Nanyuan. Finally, we have synthesized the results of the two survey methods and extracted two themes of "connections" and "development engines" for the sustainable development of the area, and we suggest strengthening the connections between the Nanyuan study area and the city

of Beijing, its connection to the Beijing central axis, and also the connections between the Nanyuan aviation cultural landscape and the ancient Nanyuan cultural landscape. We also propose to fully use the physical and cultural resources of the Nanyuan study region as a tangible and intangible growth engine.

This study contributes to the existing literature in a number of ways. First, there is a complex selection dilemma in the planning of renewal strategies for abandoned airport areas and their surroundings. We found that existing knowledge tends to be either too abstract and normative or considers the specific impacts of airports on the surrounding urban environments without accounting for differences across developmental stages and pre- versus post-abandonment. Our proposed hybrid approach emphasizes the importance of considering the varying impacts of airports on the environs across lifecycle stages and pre-/post-abandonment, to simultaneously account for the historical legacy and future development of abandoned airport sites, compensating for the deficiencies in both theoretical modeling and case studies in current research.

In addition, our study fills a gap in the historic urban landscape approach regarding the study area of abandoned airports and provides new ideas for cities around the world with similar conditions and circumstances to promote new types of living urban heritage conservation and achieve sustainable regional development.

As for the complex scenarios where it is difficult to balance the historical heritage and market demand in land use, our policy recommendations are oriented towards inspiring thought and actively seeking the most suitable balance between historical heritage and future potential based on a survey of the ancient and modern conditions, both valuing the heritage value and affirming the opportunity value for the site. Our research results fill the gap between constructive cognition and practical operation and are of great reference value in advancing the implementation of urban policies and plans, as well as helping to promote the practice of urban planning decisions in this direction.

As an urban phenomenon, change is an intrinsic aspect of the urban condition [21]. Further research is required to comprehensively and systematically refine the direction of each area's evolution in Nanyuan Airport and its surroundings, as well as its evolving historical connotations and distinctive characteristics. This research will serve as the foundation for future urban development, providing sustainable guidelines to preserve the cultural heritage and enable the creation of a world-renowned, millennia-old historical garden. At the same time, there is a need to actively experiment with mixed methods in different cultural and social contexts in order to expand their application and scope and to update them in real time to increase the understanding of the city and to provide a more reliable basis for relevant planning and policy development and updating.

**Author Contributions:** Conceptualization, Haoxian Cai; methodology, Haoxian Cai; software, Haoxian Cai; validation, Wei Duan; formal analysis, Haoxian Cai; investigation, Haoxian Cai; data curation, Haoxian Cai; writing—original draft preparation, Haoxian Cai; writing—review and editing, Haoxian Cai; visualization, Haoxian Cai; supervision, Wei Duan; project administration, Wei Duan; funding acquisition, Wei Duan. All authors have read and agreed to the published version of the manuscript.

**Funding:** This research was funded by the Fundamental Research Funds for Central Universities (2018ZY10).

**Data Availability Statement:** All data generated or analyzed during this study are included in this published article. For inquiries regarding the availability of data from this study, please contact the corresponding author.

**Acknowledgments:** The author would like to thank Wei Duan for his strong support of this research work. At the same time, he would like to thank Li Wang Wang, who enlightened his journey with her thoughtful insights. Her encouragement throughout the process of this study has been invaluable.

**Conflicts of Interest:** The authors declare no conflict of interest.

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
