# Peer review of "Mixed-Methods Approach to Land Use Renewal Strategies in and around Abandoned Airports: The Case of Beijing Nanyuan Airport"

_ijgi, doi:10.3390/ijgi12120483_

Round 1

Reviewer 1 Report

Comments and Suggestions for Authors

This paper focuses on an interesting study on Land Use Renewal in and around Abandoned Airports. Urban airports are typically large infrastructures with significant multiple impacts, thus abandoned airports are important places for long-term urban plan and update. Experiments shows that abandoned Nanyuan Airport's impact on the surrounding land use patterns is unique and significant relative to traditional planning knowledge systems in the study area. I think this study is interesting on account of its potential practical values in particular field. Through a thorough review of this work, I list some major issues in the following point-by-point.

1. The first paragraph of Introduction is so long that it is easy to lose focus for readers. In addition, the first paragraph contains a large number of related studies, and it would be appropriate to review and comment on them more in accordance with the classification clues not a long paragraph. On this basis, accurate and objective evaluation is worth receiving by readers. 

2. On line 56, it is doubt that Land use and land cover change (LUCC) can understand all of the above effects or anything else. In other words, it appears that attempts to comprehensively measure the economic, cultural and environmental impacts of airports using only LUCC are not easily accepted. The authors need to state the problem broadly and firmly. This is important because the study of the manuscript is based on this presupposition.

3. Beijing Nanyuan Airport was formally inaugurated in 1910 and officially closed in September 2019. Scarcity and complementarity of data are key challenges that the manuscript aims to overcome, thus what is the data time for DEM and OSM? 

4. Using CA-Markov simulation model for counterfactual analysis is great. However, one problem that needs to be explained clearly is how to distinguish the counterfactual difference from the learning error of the simulation model.

5. It is encouraged that the manuscript combines the two study methods of different subjects, i.e., the HUL and LUCC, to analyze typical cases. However, from the methodological perspective, the manuscript does not organically combine HUL and LUCC methods, but only analyzes some conclusions from these two aspects using different data. The fragmentation of these two methods makes this manuscript difficult to transfer and generalize in GIS disciplines. In addition, is there a data-based research method for HUL method? Neither data, methods nor conclusions have seen scalable convergence and innovation. Therefore, I believe that the current version of this manuscript is not very suitable for the International Journal of Geo-Information.

Some minor issues:

1. The uppercase or lowercase of title of papers in References should be uniform.

2. English sentences are too long to make reader tired. Especially when the author is trying to breaks it down to descript, for example ‘The research is divided into two dimensions: …’ on line 219, where the two dimensions are too separated in one sentence. Therefore, it is recommended that the author find a professional English editor or agency to make the long sentences more readable of this manuscript.

3. Generally speaking, the process framework is placed first in the Methodology, which gives readers a direct view of the methods at a global perspective.

4. The format of the manuscript needs to be carefully checked for consistency.

5. Standardize the resolution and format of figures and tables in accordance with journal requirements. The texts in many figures and tables are not legible.

Comments on the Quality of English Language

English sentences are too long to make reader tired. Especially when the author is trying to breaks it down to descript, for example ‘The research is divided into two dimensions: …’ on line 219, where the two dimensions are too separated in one sentence. Therefore, it is recommended that the author find a professional English editor or agency to make the long sentences more readable of this manuscript.

Reviewer 2 Report

Comments and Suggestions for Authors

 It is NOT clear, to me, what are the main contributions of this paper. It presents historic landscape evolution around the Nanyuan airport and uses RS and CA model to classify and simulate LUCC in modern time period. But the rationales of combing the two analyses are weak. The paper should expand intensively the discussions on how we should adopt alternative planning strategies and methods to renovate the area after the airport was shut down. 

Comments on the Quality of English Language

The paper needs thorough English editing.

Reviewer 3 Report

Comments and Suggestions for Authors

Summary

This is a well-articulated article that derives planning strategies for area around the deserted Beijing Nanyuan airport. The article digs into the ancient materials to study the landscape and land usage of the area qualitatively, and used a quantitative method to study the land use and land cover changes in recent years using remote sensing technology. However, I have questions about the methodologies and the discussions which I hope to get answer for. Please see the detailed comments and questions:

  • The article mentioned DEM data used, but I don’t seem to find anywhere mentioning the usage of the data. Would you please clarify?

  • Similar to the question above, I don’t seem to find anywhere mentioning road network data from OpenStreetMap, please clarify.

  • Line 261: ROI is first used without full name, please add the full name, and explain what it is, prefer using formula.

  • Please explain or reference Line 242-244: “In addition, remote sensing data is unable to distinguish between reserve land and cultivated land in the research region, thus reserve land is classified as cultivated land in this study”. I believe remote sensing can be used to distinguish reserved land and cultivated land, how is the conclusion drawn/where is the conclusion from?

  • Line 251-252, what is separability? What is the scale of the value?

  • The methodology for CA-Markov section is too general. Please revise to explain this part in detail. Things like mathematics behind it, the input/output to the system can be explained to help a reader with minimum knowledge of CA-Markov to understand or reproduce your study.

  • How are the discussion on ‘connection’ and ‘development engine’ tied to the results from HUL and CA-markov models? How was the understanding of the past and today of the land use in the study area used to derive the future planning strategy?

Grammar and format

  • Line 65 - 68: ‘While the Historic urban landscape (HUL) approach is also a useful 65 tool to combine urban planning with urban heritage conservation, revealing the evolu- 66 tionary patterns hidden in the historical layers and giving lessons for future sustainable 67 development [12,13]. There are also several successful instances [14,15]’ does not read right. Please revise.

  • Table 1. I think you can just drop the Number column and use the acquisition date as the index column.

  • Table 3’s font is small. Would you be able to make it bigger?

Round 2

Reviewer 2 Report

Comments and Suggestions for Authors

the revised manuscript addresses my concerns well. I think the paper is publishable.

Comments on the Quality of English Language

please make further efforts to improve the English writtng in the paper. 

Reviewer 3 Report

Comments and Suggestions for Authors

Thank authors for the detailed responses to the questions I had in the first round. Your answers fully addressed all my questions and I have no other questions.